# LEARNING FLEXIBLE GENERALIZATION IN VIDEO QUALITY ASSESSMENT BY BRINGING DEVICE AND VIEWING CONDITION DISTRIBUTIONS

## ABSTRACT

Video quality assessment (VQA) plays a critical role in optimizing video delivery systems. While numerous objective metrics have been proposed to approximate human perception, the perceived quality strongly depends on viewing conditions and display characteristics. Factors such as ambient lighting, display brightness, and resolution significantly influence the visibility of distortions. In this work, we address the question of the multu-screen quality assessment on mobile devices, as this area still tends to be undercovered. We introduce a large-scale subjective dataset collected across more than 200 Android devices, accompanied by metadata on viewing conditions and display properties. We propose a strategy for aggregated score extraction and adaptation of VQA models to device-specific quality estimation. Our results demonstrate that incorporating device and context information enables more accurate and flexible quality prediction, offering new opportunities for fine-grained optimization in streaming services. We view device and condition variability as a form of natural distributions, and our approach provides a pathway to more robust perceptual quality prediction. Ultimately, this work advances the development of perceptual quality models that bridge the gap between laboratory evaluations and the diverse conditions of real-world media consumption.

## 1 INTRODUCTION

Objective video quality assessment (VQA) is a critical task in video processing, with applications in compression, streaming, and content delivery. The goal of VQA models is to estimate human-perceived video quality using algorithmic predictions. To train and evaluate these models, researchers rely on benchmark datasets that contain subjective human scores, typically collected in controlled lab environments or through crowdsourced studies.

A key challenge in this domain is that human perception of video quality is not invariant—it varies significantly depending on viewing conditions and display characteristics, such as screen size, resolution, brightness, and ambient lighting. For example, in Barman et al. (2023), a parallel subjective test was conducted on a phone, tablet, and television. The results showed substantial differences in Mean Opinion Scores (MOS) obtained across the different devices. However, most existing datasets and VQA models either neglect these contextual factors or assume a uniform viewing environment. As a result, objective metrics that perform well in one setting may fail to generalize across diverse real-world conditions. This issue is particularly pronounced in the case of banding metrics, since banding artifacts appear differently depending on screen brightness. As shown in Safonov et al. (2024), current banding metrics demonstrate very low reliability. For the same reason, the developers of the state-of-the-art Netflix VMAF metric provide different models separately for TVs and mobile phones.

While numerous VQA models have been proposed in recent years, they typically evaluate performance on datasets with limited variability in device types and viewing environments. This limits their applicability to real-world use cases, particularly for mobile users and diverse consumer devices. While mobile devices often share similar characteristics in terms of screen size and resolution, the actual viewing conditions for mobile users can vary significantly. For instance, watching

the same video scene outdoors in bright sunlight may result in a drastically different perceptual experience compared to viewing it in a dark room. Additionally, many users deliberately reduce screen brightness to conserve battery life, further affecting visual quality. In recent years, a noticeable gap has emerged between HDR-capable and SDR-only mobile devices. Modern flagship smartphones are equipped with HDR displays capable of peak brightness levels exceeding 2500 nits, offering enhanced visibility in bright environments and more accurate rendering of dark scenes. Although the present work focuses exclusively on SDR content, a detailed analysis of how SDR and HDR content is perceived on HDR-enabled displays is provided in Ebenezer et al. (2024), highlighting additional complexities introduced by display capabilities when assessing visual quality. Without accounting for such contextual differences, VQA models risk producing misleading predictions, which can hinder user experience optimization for content providers and device manufacturers.

In this work, we address the problem of multi-screen quality assessment. First off all we collect a large-scale, crowd-sourced multi-screen video quality dataset designed to bridge this realism gap. The dataset comprises pairwise preference judgments on 200+ unique Android devices, enriched with detailed metadata: screen technology, diagonal size, peak brightness, applied brightness setting and measured ambient light. The dataset also includes pairwise comparison judgments and reference scores collected on high-resolution desktop monitors. The Figure 1 demonstrates the impact of the votes collection in different viewing conditions distributions.

On this dataset, we evaluate existing learning-based IQA and VQA models and show their limited ability to preserve correct quality orderings across different viewing conditions. To address this, we propose a training strategy and a vote aggregation framework that generalize VQA metrics for improved performance under diverse device-specific conditions. To the best of our knowledge, this is the first framework that explicitly trains quality metrics to account for viewing-device characteristics. Our findings show that incorporating device and context information substantially improves prediction accuracy and robustness across viewing conditions that can serve as a foundation for new generations of adaptive streaming solutions.

## 2 RELATED WORKS

The diversity of modern display mobile devices has important implications across many areas of video processing. However, in this work, we focus specifically on video compression, which remains one of the most fundamental and widespread applications where perceptual video quality plays a critical role. Another broad category of subjective datasets involves in-the-wild distortions, typically captured by non-professionals and characterized by artifacts such as shaking, blurring, and faded colors. These distortions are primarily related to the aesthetic aspects of content and may require separate quality assessment approaches, as shown in Wu et al. (2023). Notably, the perceived quality in such cases tends to remain relatively consistent across different display screens. Consequently, we concentrate our comparisons on compression-oriented and general quality video quality datasets and benchmarks. A summary of relevant datasets is provided in Table 1.

Numerous datasets have been proposed to support the development and evaluation of perceptual video quality metrics, particularly in the context of video compression and streaming. Large-scale JND-based datasets such as VideoSet Wang et al. (2016b) and MCL-JCV Wang et al. (2016a) enable fine-grained analysis of compression artifacts, while classic datasets like the H.264/AVC video database Nuutinen et al. (2016) provide foundational resources for benchmarking quality metrics. Recent efforts have also introduced compression-oriented benchmarks tailored to learning-based approaches, as seen in the Video Compression Dataset and Benchmark Antsiferova et al. (2022). To address streaming-specific challenges, MCL-V simulates bitrate fluctuations and stalling, and GamingVideoSet Barman et al. (2018) along with related machine learning approaches Barman et al. (2019) focus on passive gaming video quality estimation. Similarly, user-generated content (UGC) and its perceptual variability are examined in UGC-Video Li et al. (2020b), which highlights the aesthetic and artifact-rich nature of such content. Other works investigate factors influencing subjective perception beyond compression, including the impact of audiovisual interplay Seshadrinathan & Bovik (2011), temporal effects on video quality of experience Seshadrinathan et al. (2010), and Quality of Service parameters Fiedler et al. (2010). High-quality reference datasets such as the TUM HD Video Datasets Keimel et al. (2011) provide additional controlled material for model development and evaluation.

Table 1: Summary of subjective compressed video quality datasets including multi-screen and the proposed dataset.

| | Dataset | Content | Orig. | Dist. | Device number | Subjective Framework | Subj. | Ans. |
|---|---|---|---|---|---|---|---|---|
| **General** | MCL-JCV (2016) Wang et al. (2016a) | Video | 30 | 1,560 | - | In-lab | 150 | 78K |
| | VideoSet (2017) Wang et al. (2017) | Video | 220 | 45,760 | - | In-lab | 800 | - |
| | SJTU-4K (2017) Zhu et al. (2016) | Video | 20 | 200 | - | In-lab | 30 | 6K |
| | GamingVSET (2018) Barman et al. (2018) | Video | 24 | 576 | - | In-lab | 25 | - |
| | NFLX (2016) Li et al. (2016) | Video | 12 | 300 | - | In-lab | 54 | 9K |
| | KUGVD (2019) Barman et al. (2019) | Video | 6 | 144 | - | In-lab | 17 | - |
| | UGC-VIDEO (2020) Li et al. (2020b) | Video | 50 | 550 | - | In-lab | 30 | 16.5K |
| | AVT-VQDB (2019) Rao et al. (2019) | Video | 15 | 300 | - | In-lab | 50 | 15K |
| | TGV (2022) Wen et al. (2022) | Video | 150 | 1,143 | - | In-lab | 19 | - |
| | CVQAD (2022) Antsiferova et al. (2022) | Video | 36 | 1,022 | - | Crowd. | 10,800 | 320K |
| | LEHA-CVQAD (2025) Gushchin et al. (2025) | Video | 59 | 6,240 | - | Crowd. | 11,000 | 400K |
| **Multi-screen** | MSVSA (2023) Barman et al. (2023) | Video | 4 | 36 | 3 | In-lab | 26 | - |
| | MS-Banding (2024) Safonov et al. (2024) | Video | 15 | 120 | 3 | In-lab | 186 | 9,000 |
| | **Proposed** | Video | 20 | 500 | 200+ | Crowd. | 10,000 | 200K |

In Barman et al. (2023), a small-scale dataset was collected using three display devices: tablet, phone, and TV viewed in parallel. The dataset includes Mean Opinion Scores (MOS) for each device type, revealing notable differences in perceived quality across screens. In Safonov et al. (2024), the authors investigate the performance of banding metrics across three domains: TV, MacBook, and crowdsourcing platforms. To support this analysis, they introduce a dataset comprising MOS scores for 120 distorted video variants. However, the dataset in Safonov et al. (2024) is distortion-specific, with a strong emphasis on videos containing flat regions, which are particularly susceptible to banding artifacts.

Existing datasets, regardless of distortion type, typically assume controlled viewing conditions or focus on a single device type. As such, they are not well-suited for exploring the impact of device diversity and ambient conditions on perceived quality. Our work fills this gap by introducing a large-scale, multi-device dataset with associated viewing metadata, allowing for more representative evaluation of compression quality under realistic usage scenarios.

## 3 DATASET

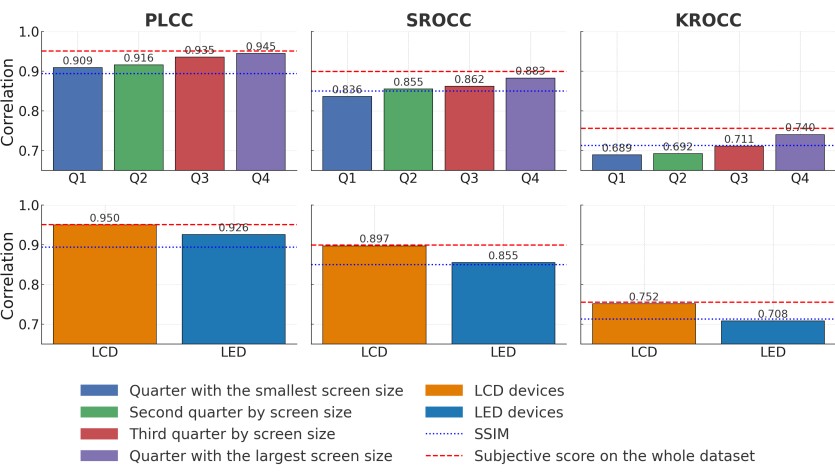

Figure 1: **Top row**: Bradley–Terry Bradley & Terry (1952) aggregated scores for subsets of devices grouped by screen size (the smallest quarter, the two middle quarters, and the largest quarter) and comparing them with aggregated scores obtained on desktop devices with large screens by Pearson (PLCC), Spearman (SROCC) and Kendall (KROCC) correlations;
**Bottom row**: correlations separately for LED and LCD displays.

## 3.1 DATASET CONSTRUCTION

We constructed a compression dedicated dataset exhibiting diverse compression artifacts. For reliable evaluation, reference videos must be of sufficiently high quality to avoid confounding recompression effects. We therefore sampled from over 18,000 high-bitrate open-source videos available on Vimeo under appropriate licenses. Only videos with a minimum bitrate of 20 Mbps were retained, resulting in a collection with an average bitrate of 130 Mbps.

To ensure representative coverage of spatial and temporal complexities, we performed clustering in the complexity space. Spatial complexity was measured as the average size of x264-encoded I-frames normalized by uncompressed frame size, while temporal complexity was defined as the ratio of average P-frame to I-frame sizes. The source videos were selected via clustering in the spatial–temporal complexity space. We introduced compression artifacts by encoding them with five encoders spanning different standards (HEVC, VVC, and AV1).

## 3.2 SUBJECTIVE TESTING

To collect pairwise preference annotations across a wide range of devices and viewing conditions, we relied on crowdsourcing. To ensure precise control over device characteristics and viewing environments, we developed an Android application that automatically recorded the device model, screen specifications, and contextual information such as brightness, ambient luminance, and orientation. Participants recruited via a crowdsourcing platform were asked to install the application, which both launched the video quality assessment interface and continuously logged relevant metadata. Screen brightness and ambient light levels were sampled every second using system APIs and the device's light sensor, when available. By default, participants used their own brightness settings; however, the application also supported enforced brightness levels. Using this feature, we collected additional subsets with brightness fixed to maximum and minimum values.

Our subjective study followed a pairwise comparison protocol, where for each source video we generated all possible pairs of its compressed versions. The reference source video itself was also included in the pool. Participants watched the pairs sequentially in full-screen mode and indicated which video exhibited better visual quality, or selected an "equal quality" option. They were allowed to replay the videos before making a choice. Each participant completed 12 comparisons, two of which were control pairs with an obvious quality difference; responses from participants who failed these checks were discarded. To improve the robustness of the results, a minimum of 15 judgments was collected for every pair. In total, the study yielded 200,000 valid annotations from nearly 10,000 contributors. Dataset parameters are summarized in Table 1. It may be noted that LCD displays correlate with the desktop scores much better than LED displays. This could be explained by the fact that most desktop monitors also use LCD matrices, which, for example, perform worse in rendering dark colors.

## 4 BLADE-CHEST MODEL

Different models can be applied to aggregate pairwise preference votes. The most commonly used approach is the Bradley–Terry Bradley & Terry (1952) model, which has been employed in large-scale datasets such as Antsiferova et al. (2022) and Gushchin et al. (2025). An alternative is the Elo rating system Elo (1978), which estimates latent scores through iterative updates after each comparison. However, such models do not account for the viewing conditions under which comparisons are made, even though these conditions can significantly influence the results. Figure 1 illustrates this effect by showing Bradley–Terry Bradley & Terry (1952) aggregated scores for subsets of devices grouped by screen size (the smallest quarter, the two middle quarters, and the largest quarter) and comparing them with aggregated scores obtained on desktop devices with large screens by Pearson (PLCC), Spearman (SROCC) and Kendall (KROCC) correlations. The correlations steadily increase from the smallest to the largest screen groups. Figure 1 also reports correlations separately for LED and LCD displays.

The Blade-Chest model Chen & Joachims (2016) leverages this limitations and makes possible to encounter conditions under which each pair were compared, as it proposes learning two vectors for each player $q_i$, namely $q_i^{\text{blade}}$ and $q_i^{\text{chest}}$. The probability of $q_i$ defeating $x_j$ is then determined by

comparing the distances between these representations: if $q_i^{\text{blade}}$ is closer to $q_j^{\text{chest}}$ than $q_j^{\text{blade}}$ is to $q_i^{\text{chest}}$, player $q_i$ is predicted to win. The use of the "blade" and "chest" embeddings provides an intuitive interpretation of the underlying model.

Therefore, we adopt the Blade–Chest model Chen & Joachims (2016) for our task, as it naturally incorporates information about the viewing conditions into the score aggregation process. To obtain rank scores from pairwise votes, we assumed that the probability of video $i$ being preferred over video $j$ in the viewing conditions $z$ is given by:

$$P(i \succ j, z) = \sigma(||f_c(q_i, z) - f_b(q_j, z)||^2 - ||f_c(q_j, z) - f_b(q_i, z)||^2) \tag{1}$$

where $q_i$ and $q_j$ are the desired subjective score estimates of videos $i$ and $j$, $\sigma(\cdot)$ denotes the sigmoid function, while $f_c(\cdot)$ and $f_b(\cdot)$ are transformation functions conditioned on the subjective score estimates and viewing conditions. These functions output $q_i^{\text{chest}}$ and $q_j^{\text{blade}}$, respectively. The functions $f_c(\cdot)$ and $f_b(\cdot)$ can take different forms; however, in this work we initialize them as fully connected neural networks, parameterized by $\theta$ and $\psi$, respectively. For the activation functions we use $\tanh$, in order to avoid purely linear behavior. Our experiments have shown that, due to the dominant influence of $q_i$ over $z$, the networks tend to degenerate into linear mappings without nonlinear activations. The vector $z$ was defined as a five-dimensional representation containing information about the display's physical size, pixel resolution, brightness, surrounding luminance, and display type.

To estimate the latent values $q_i$ and the network parameters $\theta$ and $\psi$, we employed a two-step optimization procedure based on the expectation–maximization approach. In our formulation, the latent subjective quality scores $q_i$ cannot be directly observed, while the available supervision comes only from the pairwise preference votes $\mathcal{D} = \{(i, j, z)\}$. We therefore treat $q_i$ as latent variables and optimise them jointly with the network parameters $\theta$ and $\psi$ via the Expectation–Maximisation (EM) algorithm.

The complete-data likelihood of observing a pairwise vote $(i \succ j, z)$ can be expressed as:

$$\mathcal{L}_c(q, \theta, \psi \mid \mathcal{D}) = \prod_{(i,j,z) \in \mathcal{D}} P(i \succ j, z \mid q, \theta, \psi). \tag{2}$$

Taking the logarithm, the complete-data log-likelihood becomes:

$$\log \mathcal{L}_c(q, \theta, \psi) = \sum_{(i,j,z) \in \mathcal{D}} \log \sigma \Big( ||f_c(q_i, z; \theta) - f_b(q_j, z; \psi)||^2 - ||f_c(q_j, z; \theta) - f_b(q_i, z; \psi)||^2 \Big). \tag{3}$$

Thus, the EM procedure allows us to jointly infer the latent subjective quality scores $q_i$ and optimise the transformation networks $f_c(\cdot)$ and $f_b(\cdot)$ under varying viewing conditions. A more detailed derivation of the update rules and implementation details of the optimisation procedure are provided in Appendix A.1. The obtained subjective scores exhibit a high correlation with those derived using the Bradley–Terry model on both the mobile and desktop datasets, indicating that the learned representation is consistent and logically grounded.

## 5 CONDITIONS ADAPTATION

Although we obtained subjective scores $q_i$, these values alone are of limited interest, since it is nontrivial to interpret or compare absolute score levels directly. We also trained fully connected neural networks, $f_c(\cdot)$ and $f_b(\cdot)$, which operate on score pairs; however, this formulation is not well suited for VQA applications, where it is often necessary to estimate the quality of a single video. Therefore, we treat the extraction of subjective scores as an intermediate step in adapting VQA metrics for quality prediction under varying viewing conditions.

The goal of VQA model adaptation is to enable predictions of relevant quality labels under specific viewing conditions. This is particularly useful for streaming platforms, which often optimize compression strategies for certain device types and have access to user distributions and profiles, yet still rely on standard VQA models that may under- or over-estimate the quality perceived by the end users. Modern learning-based models, both deep and traditional (e.g., VMAF), are typically trained

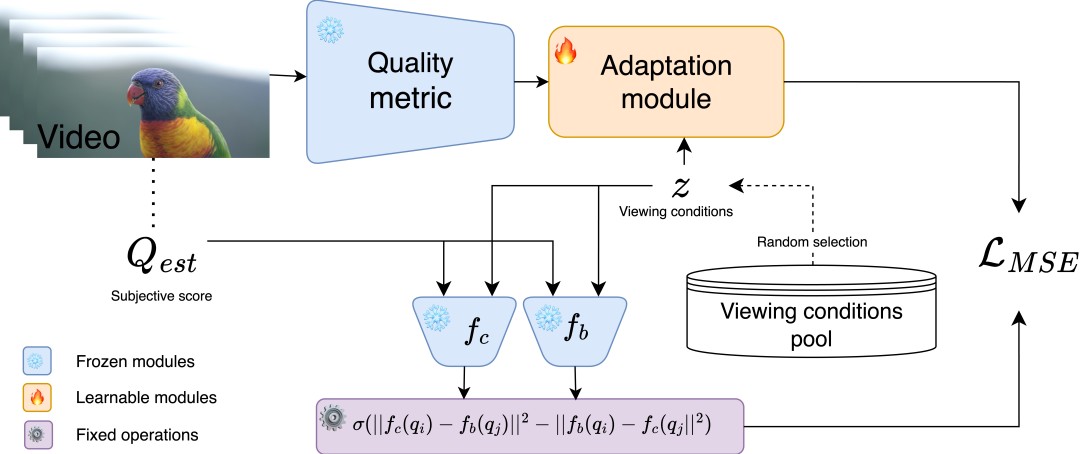

Figure 2: The training framework scheme: from the video set a random pair is sampled, and viewing conditions are selected from a distribution. The video quality metric predictions for the selected videos and the viewing conditions are processed through the adaptation module, while the estimated subjective scores together with the viewing conditions are processed through the match function using $f_c(\cdot)$ and $f_b(\cdot)$.

on large-scale datasets, so retraining them directly on our proposed dataset may be not sufficient, even when targeting condition-dependent prediction. To address this, we fine-tune the models by passing their predictions to a condition adaptation module.

The adaptation module is a lightweight fully connected neural network trained separately for each VQA model. It takes as input the model prediction together with the target viewing conditions $z$, and outputs a single value representing the predicted video quality under these conditions. The training samples are drawn from the proposed dataset. For training, the viewing conditions $z$ are generated by randomly sampling parameters from a uniform distribution, subject to hand-picked physically motivated constraints.

Now, when we have obtained subjective scores $q_i$ and trained the networks $f_c(\cdot)$ and $f_b(\cdot)$ on the real predictions, we can use their combination to expand the training set with simulated samples. Since the set of videos is fixed and new videos cannot be added without additional human judgments, we instead simulate new viewing conditions. Conditions $z$ are sampled from a uniform distribution and passed to $f_c(\cdot)$ and $f_b(\cdot)$ together with a randomly selected pair of distorted versions of the same source video. This yields the probability that video $i$ is of higher quality than video $j$ under the given conditions $z$.

So the training framework is as follows: from the video set a random pair is sampled, and viewing conditions are selected from a uniform distribution. The video quality metric predictions for the selected videos and the conditions are processed through the adaptation module, while the estimated subjective scores together with the viewing conditions are processed through the match function using $f_c(\cdot)$ and $f_b(\cdot)$. The overall framework is illustrated in Figure 5.

In most experiments, the adaptation network was implemented as a fully connected neural network of depth four, with hidden layers of size 64. We used `tanh` activations in the hidden layers to avoid linearity and applied a sigmoid activation at the output to constrain predictions to the $[0, 1]$ range.

As a result, the adaptation module is able to predict video quality based on both the metric predictions and the viewing conditions. We trained separate adaptation modules for each of the considered metrics. In addition, we conducted experiments where the adaptation module was used as a fusion mechanism across multiple metrics.

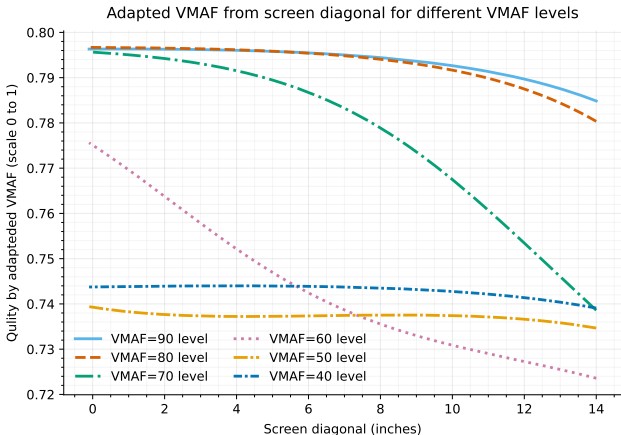

Figure 3: Relationship between the estimated quality of the VMAF adaptation module and the viewing conditions, predictions for several fixed VMAF levels with varying the screen diagonal (all other parameters are held constant).

## 6 EVALUATION

For evaluating the performance of VQA models, correlations with subjective quality scores are commonly employed. Pearson and Spearman correlations are appropriate when aggregated subjective scores (e.g., MOS, Bradley–Terry estimates) are available. However, in our dataset each crowdsourcing assessor evaluated only a small portion of the content. Consequently, the number of samples collected under identical viewing conditions is insufficient to construct reliable subjective scores for a fixed screen setup. In contrast, Kendall's rank correlation relies on the proportion of correctly ordered pairs. Since the subjective annotations in our dataset consist of pairwise preference selections under different measured viewing conditions, Kendall's rank correlation provides a natural criterion for assessing whether a VQA metric preserves the quality ordering implied by human judgments. Therefore, we adopt Kendall's rank correlation as the primary evaluation metric.

Five source videos, along with all their distorted versions, were held out as the testing set. We first applied both classical and modern neural network–based image and video quality metrics to these videos. For each participant's vote, we derived the predicted ordering from the metric outputs and then computed Kendall's rank correlation with the subjective preference. Subsequently, we trained an adaptation fully connected network separately for each of the evaluated metrics on the training portion of the dataset and also tested by Kendall's rank. Table 2 reports the Kendall rank correlations for the original metrics and their adapted counterparts.

It can be observed that even state-of-the-art metrics exhibit relatively low correlations on the raw vote data compared to the aggregated scores. This is expected, as the raw annotations are inherently noisy: for the same video under identical conditions, different participants may prefer different versions. Moreover, individual differences such as prior viewing experience or eye health can further contribute to variability. Another important factor is the viewing condition, which strongly influences perceived quality but is not explicitly modeled by existing metrics. Nevertheless, the adapted versions of the metrics substantially improve performance. While adaptation consistently enhances correlations, the remaining label noise limits performance.

Since the adaptation module employs a sigmoid activation function, the predicted scores are scaled between 0 and 1, which makes the model's outputs fairly interpretable. To illustrate the relationship between the estimated quality of the VMAF adaptation module and the viewing conditions, we plot predictions for several fixed VMAF levels while varying only the screen diagonal (all other parameters are held constant). Figure 3 shows this dependency. As expected, the perceived quality decreases as the screen size increases. It should also be noted that beyond the observed range (our dataset includes only a limited number of devices with diagonals larger than 8 inches), the predicted curves become less reliable.

Table 2: Kendall rank correlations for the original metrics and their adapted counterparts. Gain represents how score improved after metric training (red = positive, blue = negative).

|  | Metric | KROCC | KROCC (adapted) | Gain |
|---|---|---|---|---|
| Full-reference | LPIPS Zhang et al. (2018) | 0.326 | 0.555 | +0.229 |
|  | DISTS Ding et al. (2020) | 0.382 | 0.553 | +0.171 |
|  | VMAF Li et al. (2016) | 0.440 | 0.603 | +0.163 |
|  | AVQT Sodhani (2021) | 0.454 | 0.615 | +0.161 |
|  | FSIM Zhang et al. (2011) | 0.447 | 0.604 | +0.157 |
|  | SSIMULACRA Sneyers (2023) | 0.451 | 0.552 | +0.101 |
|  | HAARPSI Kastryulin et al. (2019) | 0.451 | 0.549 | +0.098 |
|  | VMAF NEG | 0.454 | 0.539 | +0.085 |
|  | VIF Sheikh & Bovik (2006) | 0.456 | 0.503 | +0.047 |
|  | PSNR | 0.448 | 0.458 | +0.010 |
|  | MS‑SSIM Wang et al. (2003) | 0.448 | 0.449 | +0.001 |
|  | SSIM | 0.454 | 0.453 | -0.001 |
|  | NLPD Laparra et al. (2017) | 0.453 | 0.450 | -0.003 |
|  | TOPIQ Chen et al. (2024) | 0.441 | 0.430 | -0.011 |
| No-reference | CLIP-IQA-PLUS Wang et al. (2023b) | 0.437 | 0.594 | +0.157 |
|  | LI2022 Li et al. (2022) | 0.393 | 0.535 | +0.142 |
|  | UNIQUE Zhang et al. (2021) | 0.424 | 0.545 | +0.121 |
|  | DBCNN Zhang et al. (2020) | 0.405 | 0.521 | +0.116 |
|  | COMPRESSED‑VQA Sun et al. (2021) | 0.404 | 0.519 | +0.115 |
|  | DOVER Wu et al. (2023) | 0.412 | 0.521 | +0.109 |
|  | VIDEVAL Tu et al. (2021) | 0.236 | 0.313 | +0.077 |
|  | MDTVSFA Li et al. (2021) | 0.426 | 0.500 | +0.074 |
|  | PAQ2PIQ Ying et al. (2020) | 0.434 | 0.505 | +0.071 |
|  | NIQE | 0.208 | 0.278 | +0.070 |
|  | RANK‑IQA Liu et al. (2017) | 0.402 | 0.464 | +0.062 |
|  | TOPIQ‑NR Chen et al. (2024) | 0.425 | 0.484 | +0.059 |
|  | KONCEPT Hosu et al. (2020) | 0.410 | 0.442 | +0.032 |
|  | BRISQUE Mittal et al. (2012) | 0.226 | 0.244 | +0.018 |
|  | LINEARITY Li et al. (2020a) | 0.419 | 0.433 | +0.014 |
|  | MUSIQ Chen et al. (2024) | 0.371 | 0.360 | -0.011 |
|  | VSFA Li et al. (2019) | 0.395 | 0.355 | -0.040 |
|  | CLIP-IQA Wang et al. (2023a) | 0.297 | 0.222 | -0.075 |

## 7 CONCLUSION

We created a new diverse dataset containing videos compressed by various encoding standards, including HEVC, AV1, and VVC, and enriched with information about labeling conditions such as screen size, screen brightness, and ambient luminance. The labels were collected on more than 200 different devices. We used this dataset to analyze how both classical and modern learning-based objective quality metrics predict video ordering across different devices. Our analysis revealed that, due to human factor annotation noise and the strong dependence of pairwise preferences on viewing conditions, existing metrics achieve only limited accuracy in predicting quality orderings. To address this limitation, we proposed a training strategy for adapting and generalizing metrics to specific viewing conditions, which results in a clear improvement in ordering quality.

The proposed dataset will be valuable for researchers and practitioners developing image- and video-quality metrics aimed at evaluating compression artifacts and optimizing solutions for diverse devices. It can be used to train models that assess video compression quality with higher accuracy and viewing condition specific precision, bringing more flexible encoders optimization. In future work, we plan to further expand the dataset by increasing the number of original source videos, incorporating additional encoders and devices.

## 8 LIMITATIONS

In this work, we did not retrain the evaluated metrics on the proposed dataset; instead, we applied the adaptation module to the outputs of pre-trained models without tuning their internal parameters. While this approach demonstrates the feasibility of condition-aware adaptation, it may limit the full potential of the underlying metrics. Future work will therefore include retraining or fine-tuning metrics directly on subsets of the dataset to achieve further improvements. Another limitation is that, in the current design, the adaptation module must be inferences separately for each target device, which may be inefficient when scaling to a large number of devices. As a next step, we plan to explore direct mappings from device distributions to score distributions, enabling more efficient and unified adaptation across diverse viewing conditions.

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

# A APPENDIX

## A.1 OPTIMISATION PROCEDURE

In this appendix, we provide a detailed derivation of the Expectation–Maximisation (EM) procedure used to jointly infer the latent subjective scores $\{q_i\}$ and optimise the parameters of the transformation networks $f_c(\cdot)$ and $f_b(\cdot)$.

### A.1.1 COMPLETE-DATA LIKELIHOOD

Let $\mathcal{D} = \{(i, j, z)\}$ denote the set of observed pairwise preference votes under viewing condition $z$, where $(i \succ j, z)$ indicates that video $i$ was preferred over video $j$. The probability of this observation under our model is

$$P(i \succ j, z \mid q, \theta, \psi) = \sigma\left(\|f_c(q_i, z; \theta) - f_b(q_j, z; \psi)\|^2 - \|f_c(q_j, z; \theta) - f_b(q_i, z; \psi)\|^2\right), \quad (4)$$

where $\sigma(\cdot)$ is the sigmoid function, and $\theta$ and $\psi$ are the parameters of $f_c$ and $f_b$, respectively.

The complete-data likelihood is then

$$\mathcal{L}_c(q, \theta, \psi \mid \mathcal{D}) = \prod_{(i,j,z)\in\mathcal{D}} P(i \succ j, z \mid q, \theta, \psi), \quad (5)$$

and the corresponding log-likelihood is

$$\log \mathcal{L}_c(q, \theta, \psi) = \sum_{(i,j,z)\in\mathcal{D}} \log \sigma\Big(\Delta_{ij}(z; q, \theta, \psi)\Big), \quad (6)$$

where

$$\Delta_{ij}(z; q, \theta, \psi) = \|f_c(q_i, z; \theta) - f_b(q_j, z; \psi)\|^2 - \|f_c(q_j, z; \theta) - f_b(q_i, z; \psi)\|^2. \quad (7)$$

### A.1.2 E-STEP

In the E-step, we compute the expected log-likelihood with respect to the posterior of the latent variables $q$, given the current parameter estimates $(\theta^{(t)}, \psi^{(t)})$:

$$Q(q, \theta, \psi \mid \theta^{(t)}, \psi^{(t)}) = \mathbb{E}_{q|\mathcal{D},\theta^{(t)},\psi^{(t)}}\left[\log \mathcal{L}_c(q, \theta, \psi)\right]. \quad (8)$$

In practice, this expectation is approximated by point estimates of the latent scores $\{q_i\}$ obtained from the previous iteration, i.e. we set

$$q^{(t)} = \arg\max_q \log \mathcal{L}_c(q, \theta^{(t)}, \psi^{(t)}). \quad (9)$$

### A.1.3 M-STEP

In the M-step, we maximise the surrogate function $Q$ with respect to both the latent scores and network parameters:

$$(q^{(t+1)}, \theta^{(t+1)}, \psi^{(t+1)}) = \arg\max_{q,\theta,\psi} Q(q, \theta, \psi \mid \theta^{(t)}, \psi^{(t)}). \quad (10)$$

This reduces to gradient-based optimisation of the log-likelihood. Specifically, the gradients are given by

$$\nabla_{q_i} \log \mathcal{L}_c = \sum_{(i,j,z)\in\mathcal{D}} (1 - \sigma(\Delta_{ij})) \, \nabla_{q_i} \Delta_{ij}(z; q, \theta, \psi), \quad (11)$$

$$\nabla_{\theta} \log \mathcal{L}_c = \sum_{(i,j,z)\in\mathcal{D}} (1 - \sigma(\Delta_{ij})) \, \nabla_{\theta} \Delta_{ij}(z; q, \theta, \psi), \quad (12)$$

$$\nabla_{\psi} \log \mathcal{L}_c = \sum_{(i,j,z)\in\mathcal{D}} (1 - \sigma(\Delta_{ij})) \, \nabla_{\psi} \Delta_{ij}(z; q, \theta, \psi). \quad (13)$$

The terms $\nabla_{q_i}\Delta_{ij}$, $\nabla_{\theta}\Delta_{ij}$, and $\nabla_{\psi}\Delta_{ij}$ can be computed via automatic differentiation since $f_c(\cdot)$ and $f_b(\cdot)$ are implemented as neural networks.

### A.1.4    REGULARISATION AND IDENTIFIABILITY

Since the latent scores $\{q_i\}$ are only identifiable up to affine transformations, we impose constraints to avoid degeneracy:

$$\frac{1}{N} \sum_{i=1}^{N} q_i = 0, \qquad \frac{1}{N} \sum_{i=1}^{N} q_i^2 = 1. \tag{14}$$

These constraints normalise the latent quality scale, ensuring that the scores are comparable across training runs.

### A.1.5    SUMMARY

The EM optimisation alternates between:

1. Updating the latent quality scores $\{q_i\}$ based on the current network parameters (E-step),
2. Updating the network parameters $(\theta, \psi)$ by maximising the surrogate log-likelihood (M-step),

until convergence. In practice, we implement both steps jointly using stochastic gradient descent, with normalisation of $\{q_i\}$ applied after each iteration to enforce identifiability.

