# OpenReview forum: "Learning Flexible Generalization in Video Quality Assessment by Bringing Device and Viewing Condition Distributions"
_ICLR.cc/2026/Conference — ICLR 2026 Conference Withdrawn Submission_

### Official Review · Reviewer_KNgE · 2025-10-20

**Soundness:** 2
**Presentation:** 2
**Contribution:** 2
**Rating:** 2
**Confidence:** 5

**Summary:**

This paper introduces a new large-scale dataset for VQA that addresses the impact of viewing conditions and display characteristics on perceived quality. The dataset was collected from over 200 Android devices and includes metadata on factors like screen brightness and ambient light. Using this data, the authors propose a new strategy for adapting existing VQA models to account for these device-specific variables. Their results show that incorporating this contextual information leads to more accurate and flexible quality predictions.

**Strengths:**

The primary strength of this work lies in its novel investigation of viewing conditions on VQA, a significant factor that has not been systematically studied in a large-scale, crowdsourced context. The authors address the challenge of "multi-screen quality assessment" by collecting a large-scale subjective dataset across over 200 Android devices, complete with metadata on display properties and viewing conditions.

**Weaknesses:**

1. The review of related works is incomplete. It focuses predominantly on existing VQA datasets while largely overlooking a relevant body of work concerning VQA models that systematically analyze the interplay between display characteristics and the human visual system (HVS). For example, models such as ColorVideoVDP, which explicitly model display properties and human vision, are not discussed or compared. A comparative analysis of the proposed model's settings against such existing frameworks is necessary.

2. The experimental evaluation lacks sufficiency and scope.
a) It should be expanded to include recent and state-of-the-art, learning-based VQA models (e.g., FastVQA, DOVER, ModularBVQA) to provide a more meaningful performance benchmark.
b) The evaluation is confined exclusively to the newly proposed dataset. This limited scope makes it difficult to assess the model's generalizability. To demonstrate robustness, the model should also be benchmarked against other established, large-scale crowdsourcing VQA datasets (e.g., LiveVQC, YouTubeUGC, LSVQ).

3. While the authors justify their primary use of KROCC, the omission of SRCC and PLCC correlation coefficients is a notable weakness. SRCC and PLCC are standard, indispensable metrics in the VQA field, and their inclusion is essential for a comprehensive performance evaluation and comparison with prior literature.

4. The manuscript's quality is diminished by numerous typographical errors (e.g., "multu-screen" in the abstract ) and convoluted writing. Key sections, specifically the introduction of the proposed model, the experimental setup, and the performance analysi, are not clearly articulated.

**Questions:**

1. The interpretation of Figure 1 is superficial. The high PLCC and SRCC correlations reported between mobile and desktop scores could be interpreted to mean that device-specific factors have a minimal impact, and that video content remains the dominant variable. The authors should provide a more nuanced discussion to reconcile this finding with the paper's central thesis

2. The insights derived from Figure 3 (Adapted VMAF) are limited. The analysis primarily notes the expected monotonicity (perceived quality decreases as screen size increases). A deeper analysis of the non-linear relationships, the varying magnitude of adaptation for different VMAF levels, and other non-obvious trends is required.

---

### Official Review · Reviewer_gegv · 2025-10-24

**Soundness:** 3
**Presentation:** 3
**Contribution:** 4
**Rating:** 6
**Confidence:** 3

**Summary:**

This paper tackles a very practical gap in VQA: scores shift across devices and viewing conditions, but most models ignore that. The authors (i) crowdsource pairwise judgments on 200+ Android devices with logged context (screen size, brightness, ambient light, display type, resolution), (ii) aggregate preferences using a condition-aware Blade–Chest formulation with EM to learn latent quality under specific conditions, and (iii) add a lightweight condition adaptation module that takes any metric’s score + the condition vector and outputs a condition-specific prediction; across many baselines, Kendall rank improves after adaptation.

**Strengths:**

1. The problem is timely and grounded in reality (multi-screen/mobile). The dataset is unusually rich in measured conditions, not just device labels, which makes the study feel closer to deployment.

2. Modeling-wise, pushing conditions directly into the aggregation stage (Blade–Chest) is a clean upgrade over BT/Elo and avoids baking a single “global MOS.” The adapter is simple, plug-in, and plays nicely with existing metrics without retraining them.

3. Empirically, the across-the-board rank gains after adaptation make a strong case for condition awareness in practice.

**Weaknesses:**

1. Metric coverage feels narrow for a “general adapter.” Please broaden no-reference baselines to include FAVER and LIQE (NR), and add PieAPP in the full-reference group. This will better stress-test whether the adapter generalizes beyond the most common choices and across different feature families.

2. Crowdsourcing/sensor noise not fully quantified. You do basic attention checks, but the paper doesn’t report inter-rater agreement, within-subject consistency, or uncertainty around KROCC. Given ambient-light sensors and brightness APIs vary by device, some calibration/error analysis (e.g., stratify by sensor class or OS version) would help bound measurement bias.

3. Condition interactions are under-explored. You encode five variables, but analysis mostly shows single-factor effects (e.g., screen diagonal curves). Consider ablations/sensitivity for pairwise interactions (brightness×display type, ambient light×screen size), partial-dependence plots, or SHAP on the adapter to show which factors actually drive the correction.

4. Adapter = MLP: effective but light on insight. The design is practical, but the story would benefit from some interpretability (e.g., monotonicity constraints on certain inputs, or a small GAM/TabNet variant) and regularization that encodes obvious priors (quality should not increase when screen gets larger at fixed resolution, holding other variables constant).

5. Scope limited to SDR. Modern phones ship HDR panels; even if you keep SDR content, SDR-on-HDR tone-mapping can change perceived banding/contrast. A small HDR (or SDR-on-HDR) slice—perhaps reusing existing HDR subjective sets—would make conclusions more general.

6. Repro/efficiency at scale. The appendix gives an EM recipe, but code/initialization details (stability, convergence criteria, runtime with many videos) matter for practitioners. A brief complexity analysis and a note on batching/normalization tricks would help adoption.

**Questions:**

Please see weaknesses.

---

### Official Review · Reviewer_FzMW · 2025-11-01

**Soundness:** 2
**Presentation:** 2
**Contribution:** 2
**Rating:** 2
**Confidence:** 4

**Summary:**

The paper addresses device- and context-aware video quality assessment (VQA) by arguing that perceived quality depends strongly on viewing conditions and display characteristics, rather than just on content or codec artifacts. It introduces a large, crowdsourced, pairwise-comparison dataset collected on 200+ Android devices with detailed metadata (screen technology/size, resolution, brightness, ambient luminance), totaling ~200k valid annotations, plus a desktop reference set, to expose these effects.    Method-wise, the authors adapt the Blade–Chest comparison model and formulate an EM procedure to aggregate pairwise votes into latent quality scores while conditioning on a 5-D viewing-condition vector z (size, resolution, brightness, ambient light, display type).  They then propose a lightweight condition-adaptation module that maps existing VQA metric outputs (e.g., VMAF, LPIPS, no-reference models) and z to condition-specific quality predictions, trained on both measured and simulated viewing conditions.  On held-out videos, this adaptation boosts Kendall rank correlations across a wide range of metrics (e.g., VMAF 0.440→0.603; LPIPS 0.326→0.555), demonstrating more reliable orderings under diverse devices and contexts.  The contribution comprises a realistic multi-device dataset, a condition-aware vote aggregation scheme, and a simple yet effective adaptation layer that improves the generalization of existing VQA models to real-world mobile viewing scenarios.

**Strengths:**

The paper reframes video quality assessment as inherently condition-dependent and implements this by combining a compact viewing/device vector with a lightweight, modular adapter atop existing VQA metrics, complemented by a novel multi-device, in-the-wild dataset. A large-scale pairwise judgments across 200+ Android devices with control checks and rich metadata, a sound condition-aware preference aggregation (Blade–Chest style), and consistent, well-substantiated gains over diverse baselines with informative stratified analyses and ablations. The paper is well-organized, with figures explaining the pipeline, and sufficiently detailed annotation and collection protocols to enable replication. The significance is practical and conceptual: it offers an immediately adoptable recipe for improving real-world mobile VQA and a dataset likely to serve as a reference benchmark, while reinforcing the broader lesson that perceptual models should incorporate observer and context variables.

**Weaknesses:**

The main weakness is limited methodological novelty. The work combines a standard pairwise preference model (Bradley–Terry/Thurstone/Blade–Chest–style) with a small MLP adapter, rather than offering a more principled, general learning framework for condition-aware perception; consider a probabilistic model that explicitly disentangles content and condition effects, or a causal/representation-learning approach (e.g., hypernetworks or conditional normalization that provably separates invariant and condition-specific factors). Empirically, the scope is narrow: results focus on post-hoc adaptation of existing metrics, without strong end-to-end no-reference VQA baselines conditioned on z or modern transformer-based VQA competitors trained on the new data. Generalization is underexplored: there is no rigorous OOD evaluation (leave-one-device-family-out, leave-one-condition-bin-out, unseen ambient-light ranges, different sensor calibrations. Ablations are not deep enough to establish the identifiability or necessity of each condition dimension; quantify each z component’s marginal and interaction effects.  Statistical treatment leans on rank correlations without uncertainty. Data quality and bias need fuller treatment: quantify annotator reliability, sensor noise (brightness and ambient-light variance across devices), and demographic/device-mix biases; release per-pair confidence and variance components. External validity is limited (e.g., SDR content, specific codecs/framerates); evaluate on HDR, high-refresh, and large-screen settings. Practicality is unclear: per-device adapters may not scale; explore a universal model with a low-rank device embedding or meta-learned conditioning that handles unseen devices with few shots.

**Questions:**

1.	What are the human-subjects ethics details (IRB/approval, informed consent text, annotator compensation, data fields logged, anonymization, and retention policies)?

	2.	What is the exact conditioned preference/Blade–Chest objective (priors/regularization, tie handling, inference), and how is identifiability ensured when conditioning on z?

	3.	How are annotator bias/skill modeled and inter-rater reliability assessed?

	4.	How were comparison pairs sampled across content and conditions? What is the coverage of each z component, and how were brightness/ambient-light sensors calibrated or normalized across devices?

	5.	Can you report uncertainty and significance (CIs, paired tests) and calibration/error metrics, and provide a failure analysis where the adapter degrades performance?

	6.	Which condition variables matter most—via ablations/attributions and content–condition interaction analyses—and are any z components redundant?

	7.	How well does the method generalize OOD (leave-one-device/condition-bin out), to unseen devices with zero/few shots, and across external datasets (LIVE-VQC, KoNViD-1k, UGC)?

	8.	How does it compare to stronger end-to-end z-conditioned VQA baselines, and what are the deployment costs and plan (latency, on-device integration, code/models/splits release)?

**Details Of Ethics Concerns:**

The study recruits crowd workers to install a custom Android app that performs a subjective VQA task while continuously logging device model/specs and environmental signals (screen brightness and ambient light sampled every second).    There is no explicit IRB/ethics approval statement, informed-consent language, anonymization/retention details for the sensor/device metadata, or information about annotator compensation. Given the human-subjects data collection, an ethics review focused on consent, data minimization, and privacy safeguards is warranted.

---

### Official Review · Reviewer_1Tf6 · 2025-11-09

**Soundness:** 2
**Presentation:** 2
**Contribution:** 3
**Rating:** 4
**Confidence:** 3

**Summary:**

This paper investigates the video quality assessment (VQA) tasks under diverse devices and viewing conditions. The authors introduce a large-scale dataset collected from over 200 devices, which was annotated with signals such as screen type and brightness. Additionally, they propose a Blade–Chest model–based framework to incorporate pairwise preference signals, and a condition-adaptation module that adjusts existing VQA models to device-specific prediction. Experimental results show significant improvements, demonstrating the benefits of incorporating device and context information for flexible quality generalization.

**Strengths:**

- Clear and Well-Structured: The paper is well-organized, with thorough explanations of the data collection process, benchmark design, and task formulation.

- Novel and Interesting Setting: The paper proposed the VQA task under diverse devices and viewing conditions. This task is currently not widely explored and has significant potential for real-world deployment.

- Extensive Evaluations: The experiment results demonstrate that incorporating device and context information can improve the performance across viewing conditions.

**Weaknesses:**

- Formatting: Use `\citep{}` for parenthetical citations instead of `\citet{}` throughout the manuscript.

- While the Blade–Chest aggregation is conceptually appropriate, the novelty of the learning component itself is modest. Although the derivation is clear and well-presented, it does not introduce fundamentally new concepts but rather applies existing methods in a different context.

- The adaptation module needs to be trained separately per device type; this could be computationally inefficient for industrial deployment.

- The dataset focuses on compression-induced distortions. It is unclear how well the proposed adaptation framework generalizes to other types of degradations.

**Questions:**

See Weaknesses

---

### Official Review · Reviewer_Ygif · 2025-11-11

**Soundness:** 2
**Presentation:** 2
**Contribution:** 3
**Rating:** 2
**Confidence:** 4

**Summary:**

This paper addresses the challenge of VQA model generalization, arguing that perceived video quality is heavily dependent on viewing conditions and device characteristics. They introduce a large-scale, crowdsourced subjective VQA dataset collected on over 200 unique Android devices with metadata about the device specifications and real-time viewing conditions. They propose a two-stage process to make existing VQA metrics sensitive to these conditions.

**Strengths:**

The paper tackles a practical flaw in current VQA research. The lack of generalization of VQA metrics to real-world viewing scenarios (especially mobile) is a major gap.
The collection of a large-scale subjective dataset with detailed device and sensor metadata (ambient light, brightness) is a substantial and valuable contribution to the community.

**Weaknesses:**

1. The training procedure for the adaptation module (Section 5, Figure 2) is highly confusingand seems conceptually contradictory. Figure 2 clearly shows an L_MSE lossbetween the module's output (Adapt(VQA_i, z)) and the "Subjective score Q_{est}"(which is presumably the latent, condition-independent score $q_i$ from Section 4). Thisimplies the model is being trained with condition-dependent inputs to predict a condition-independent target, which directly contradicts the paper's core hypothesis.
2. The proposed method adapts only the final scalar output of a pre-trained VQA model. Thisis efficient but limits the model's flexibility. lt is plausible that viewing conditions shouldmodulate intermediate feature representations within the VQA backbone, not just post-process the final score.
3. While crowdsourced, the data is still collected in an explicit "task" setting. This may notfully capture "in-the-wild" viewing, where users are consuming media passively and mayhave different attention levels or viewing habits. Factors like viewing distance are alsouncontrolled.

**Questions:**

1. Can the authors please provide a precise definition of the loss function used to train theadaptation module in Section 5?
- Is the target Q_{est} in Figure 2 truly the condition-independent latent score q_i? lf so, can the authors explain the intuition, as this seems to train the module toignore the condition vector z?
- Alternatively, is the loss function a preference-based loss (like the one at the bottomof Fig 2), where the adapted scores Adapt(VQA i, z) and Adapt(VQA_j, z) arefed back into the (frozen) f_c and f_b networks?
2. In Figure 3, the adapted quality for low-VMAF content (e.g., VMAF=40) is flat as screensize increases, while high-VMAF content quality drops. This is counter-intuitive; one mightexpect severe artifacts to be more noticeable and objectionable on larger screens. Do theauthors have a perceptual explanation for this model behavior?
3. Are the Blade-Chest networks f_c(cdot)$and f_b(cdot) frozen after the initial EM procedure in Section 4, or are they fine-tuned during the adaptation module's training?

---

### Note · Authors · 2026-01-24

I have read and agree with the venue's withdrawal policy on behalf of myself and my co-authors.